# 3-Aryl-4-nitrobenzothiochromans *S,S*-dioxide: From Calcium-Channel Modulators Properties to Multidrug-Resistance Reverting Activity

**DOI:** 10.3390/molecules25051056

**Published:** 2020-02-27

**Authors:** Matteo Micucci, Maurizio Viale, Alberto Chiarini, Domenico Spinelli, Maria Frosini, Cinzia Tavani, Massimo Maccagno, Lara Bianchi, Rosaria Gangemi, Roberta Budriesi

**Affiliations:** 1Dipartimento di Farmacia & Biotecnologia, Alma Mater Studiorum-University of Bologna, Via Belmeloro 6, 40126 Bologna, Italy; matteo.micucci2@unibo.it (M.M.); alberto.chiarini@unibo.it (A.C.); 2IRCCS Ospedale Policlinico San Martino, U.O. Bioterapie, L.go R. Benzi 10, 16132 Genova, Italy; maurizio.viale@hsanmartino.it (M.V.); rosaria.gangemi@hsanmartino.it (R.G.); 3Dipartimento di Chimica “G. Ciamician”, Alma Mater Studiorum-University of Bologna, via F. Selmi 2, 40126 Bologna, Italy; domenico.spinelli@unibo.it; 4Dipartimento di Scienze della Vita, Università degli Studi di Siena, Via A. Moro 2, 53100 Siena, Italy; maria.frosini@unisi.it; 5Dipartimento di Chimica e Chimica Industriale, Università degli Studi di Genova, Via Dodecaneso 31, 16146 Genova, Italy; cinzia.tavani@unige.it (C.T.); massimo.maccagno@unige.it (M.M.); lara.bianchi@unige.it (L.B.)

**Keywords:** 3-aryl-4-nitrothiochromans *S*,*S*-dioxide, l-Type Calcium Channels (LTCC), anticancer therapy, multidrug resistance (MDR1), in vitro experiments, cardiovascular activity

## Abstract

Our research groups have been involved for many years in studies aimed at identifying new active organic compounds endowed with pharmacological properties. In this work, we focused our attention on the evaluation of cardiovascular and molecular drug resistance (MDR) reverting activities of some nitrosubstituted sulphur-containing heterocycles. Firstly, we have examined the effects of 4-nitro-3-(4-methylphenyl)-3,6-dihydro-2*H*-thiopyran *S,S*-dioxide **5**, and have observed no activity. Then we have extended our investigation to the 3-aryl-4-nitrobenzothiochromans *S,S*-dioxide **6** and **7**, and have observed an interesting biological profile. Cardiovascular activities were assessed for all compounds using ex vivo studies, while the MDR reverting effect was evaluated only for selected compounds using tumor cell lines. All compounds were shown to affect cardiovascular parameters. Compound **7i** exerted the most effect on negative inotropic activity, while **6d** and **6f** could be interesting molecules for the development of more active ABCB1 inhibitors. Both **6** and **7** represent structures of large possible biological interest, providing a scaffold for the identification of new ABCB1 inhibitors.

## 1. Introduction

Cancer chemotherapy, as in the case of antimicrobial chemotherapy, has had trouble with resistance to treatment. This phenomenon occurs mainly through molecular mechanisms that may be affected by novel compounds that may be associated to cancer chemotherapy [1]. Nowadays, cancer treatment based on a combination of drugs with different mechanisms of action seems to be the best strategy to avoid multi drug resistance (MDR) [2,3]. This approach has proved to be successful for some kinds of cancer such as leukemias, while for other types of high-spread cancers, including breast and lung cancer, the failure rate is still very high.

Studies dating back to the second half of the last century identified the presence of a P-glycoprotein (Pgp) that is correlated to the onset of drug resistance [4,5]. The discovery that the high concentration of Pgp in resistant cells is independent of the chemical class of drugs used enlightens a further complicating factor. The mechanisms underlying multi drug resistance (MDR) are not fully understood, and they are responsible for the decrease of intracellular drugs concentration [6]. To counteract this phenomenon, various strategies, including the search for chemical entities able to inhibit Pgp, have been investigated.

Calcium channels might also be targets for pathologies other than cardiovascular diseases. Many calcium channel blockers of different structural classes, including diltiazem-like drugs, in fact, represent modulators of MDR (resistance modifiers, chemosensitizers). Since the sensitized tumor cells do not express voltage-gated calcium channels, the chemosensitization exerted by these drugs is independent on their activity as intracellular calcium-regulators. Unfortunately, most of them affect cardiovascular parameters at the concentrations required for a successful reversal of MDR, preventing its clinical use. Thus, novel compounds endowed with low calcium channel blocking activity may be considered in the search for relatively specific MDR reverting agents.

Our research group has been involved for a long time in a research program aimed at finding new l-Type calcium channel modulators for peripheral or central application [7,8] by modifying the chemical structure to select alternative action for the cardiovascular system.

In the frame of our research, we examined the activity of a small library (21 compounds) of variously substituted 8-aryl-8-hydroxy-8*H*-[1,4]thiazino[3,4-*c*][1,2,4]oxadiazol-3-ones (**1**) as calcium-entry blockers by means of functional studies on isolated guinea-pig left and right atria and K^+^-depolarized aortic strips [9]. This new class of compounds is structurally correlated to diltiazem (**2**) [10] and to pyrrolobenzothiazines (**3**) [11], which are well known as potent calcium antagonists with a strong selectivity for cardiac tissues over vascular ones, and thus widely used as therapeutic agents in cardiovascular diseases. Interestingly, we observed [9] that at least two of the examined compounds (**4a**: X = Cl; **4b**: X = Br) are more potent than diltiazem itself as l-type calcium channel modulators, and they selectively affect the heart contractility without effect on vascular tissues (Figure 1).

Following the identification of this new chemotype that is able to selectively bind to l-type calcium channels, we explored the possibility for this class of compounds to exhibit MDR reverting properties, such as the parent compound diltiazem [12], independent of the action on l-type calcium channels even if the resistant cells have high levels of intracellular calcium accumulation [13]. For this reason, some selected oxadiazolones have been studied on doxorubicin-resistant cell lines. Some of these compounds can increase accumulation of doxorubicin in the ovarian carcinoma multidrug resistant cell line A2780 through inhibition of MDR1 [14].

Considering the calcium-entry blocker (CEB) activity and the chemical structures of the diltiazem related compounds purposely designed and studied [7,9,15], we focused our attention on a new class of compounds, namely 4-nitrothiopyran *S,S*-dioxide (**5**) [16] and on the benzocondensed analogues (viz. the thiochromans *S,S*-dioxide **6** and **7**) [17,18] (Figure 2).

In the latter chemotypes, the possibility of different type/degrees of substitution at C-2 and/or C-3 of the thiopyran ring should expand the opportunity for a significant biological activity from l-type calcium channel antagonism to MDR reverting activity. Relevant results on the cardiovascular characterization of the new compounds are reported hereinafter. In addition, we also investigated the effects on MDR reverting activity of some selected compounds to identify new chemical entities with this peculiarity.

## 2. Results

### 2.1. Chemistry

The synthetic procedure for **5** (Scheme 1 [16]) and for the benzocondensed derivatives **6a**–**d**, **f**–**j** and **7a**–**e**, **g**–**l**, isolated as diastereomeric mixtures, with a generally high degree of diastereoselectivity (Scheme 2 [17,18]), features an overall ring-opening/ring-closing protocol starting from 3-nitrothiophene (**8**) and 3-nitrobenzo[*b*]thiophene (**9**), respectively. It encompasses (a) substrate ring-opening with pyrrolidine/AgNO_3_, (b) trapping of the intermediate silver sulfide with a suitable halide, (c) replacement of the pyrrolidino moiety with the aryl of a Grignard reagent, (d) oxidation of sulfur, and (e) cyclization to the final thiopyran (**5**) or thiochromans (**6** or **7**) dioxide. The procedure is based on a long-standing project on the synthetic exploitation [16,18,19,20,21] of the valuable polyfunctionalized nitrobutadienic building-blocks, which originate from the non-benzenoid behavior of nitrothiophenes, viz. the ring-opening reaction that follows the nucleophilic attack of primary or secondary amines.

As a matter of fact, the 5- to 6-member ring-expansion protocol had been earlier applied to the synthesis of 4-nitro-3-(4-methylphenyl)-3,6-dihydro-2*H*-thiopyran *S,S*-dioxide **5** [16], revealing for this heterocycle a thermal instability, which possibly contributes to the disappointing preliminary results on its pharmacological activity (see below in the text). This outcome has led us to turn our attention to the benzocondensed derivatives, on the grounds of both the experimental higher stability of 3-aryl-4-nitrothiocromans *S*,*S*-dioxide (**6**) and 3-aryl-4-nitro-2-phenylthiochromans *S*,*S*-dioxide (**7**) vs. **5** and the results of some virtual-screening investigations.

This fact would most likely lead to an improved activity on shifting from **5** to **6** or **7**. Furthermore, the encouraging results on the activity of **6** (to be discussed below) have fostered our interest in a deeper evaluation of this class of sulfur heterocycles as potential cardioregulators. Thus, the thiochromans **7** have been designed as likely suitable candidates for the programmed screening as far as (a) the phenyl at C-2 should conceivably alter the geometric array of the molecule, and (b) different and/or variously-substituted aryl/heteroaryl moieties can be introduced at either C-2 (by choosing a proper trapping halide for the silver thiolate in step (b) of Scheme 2), C-3 (by choosing a proper Grignard reagent for step (c)) or at both; this “diversity” should of course increase the probability of highlighting more effective drugs.

From the stereochemical point of view, in the case of compounds **7,** the situation was expected to be definitely more complex because of the presence of the additional stereocenter at C-2, which is generated together with those at C-3 and C-4 during the cyclization step (e). As a matter of fact, interestingly enough, a significant stereoselectivity occurs, strongly favoring the *cis,trans*-racemate **7** (Figure 3) (the assignment of structure being based on ^1^H NMR data), which is an aspect that is conceivably related to the crowding of the molecule and that has been already discussed in detail [18].

### 2.2. Ex Vivo Functional Studies

#### 2.2.1. Cardiac Activity

The cardiac profile of all new compounds (**5**, **6a**–**d**, **f**–**j**, and **7a**–**e**, **g**–**l**) was investigated using guinea-pig isolated left and right atria to evaluate their inotropic and/or chronotropic effects, respectively. In particular, the percent decrease of developed tension on isolated left atrium driven at 1 Hz and on the spontaneously beating right atrium (negative inotropic activity), and the percent decrease in atrial rate on spontaneously beating right atrium (negative chronotropic activity), were checked at increasing concentration.

Thiopyran **5** is the only compound lacking cardiac action; in fact, it has no negative chronotropic activity on the spontaneously beating right atrium and no negative inotropic activity either on the left atrium driven at 1 Hz or on the right atrium spontaneously beating. All the other compounds are also devoid of bradycardic effects while all have a negative inotropic action both on the left atrium and on the right atrium. Thiochromans **6a**, **7e**, **7g**, **6h**, **7i**, and **7l** are, respectively, about 1.8, 4.2, 1.6, 2.5, 4.6, and 2.5 times more potent than diltiazem. Compounds **6d**, **6f**, **6g**, and **6i** are less potent than diltiazem 1.7, 4.5, 2.6, and 1.8 times, respectively; while compounds **7a**, **6b**, **7b**, **6c**, **7c**, **7d**, **7h**, and **7j** have potency comparable to that of diltiazem (Table 1). The most potent compounds on this parameter are **7i** and **7e** (EC_50_ = 0.17 µM (c.l. 0.12–0.21); EC_50_ = 0.19 µM (c.l. 0.14–0.26), respectively). In the same way, all the tested compounds, with the exception of **5**, have a negative inotropic activity even on the spontaneously beating right atrium. The most potent compounds on this parameter are **6f** and **7j** (EC_50_ = 0.28 µM (c.l. 0.17–0.38); EC_50_ = 0.19 µM (c.l. 0.13–0.28), respectively). For diltiazem, this activity has not been evaluated because it has a bradycardiac activity, which affects the inotropic effect.

#### 2.2.2. Smooth Muscle Spasmolytic Activity

All compounds were tested on K^+^-depolarized (80 mM) guinea-pig vascular (aortic strips) smooth muscle to assess calcium antagonist activity. Some selected compounds were tested also on guinea pig ileum to test the activity on K^+^-depolarized (80 mM) non-vascular smooth muscle. Compounds were checked at increasing concentration to evaluate the percent inhibition of calcium-induced contraction on K^+^-depolarized vascular (aortic strips) and non-vascular (ileum) smooth muscle. Data are presented in Table 2 together with those for diltiazem.

None of the studied compounds had spasmolytic effects on K^+^-depolarized vascular smooth muscle in contrast to diltiazem, which, in analogy with the other classes of calcium antagonists, has an intrinsic activity greater than 50% and μM potency (Table 2).

Some compounds, selected on the basis of their chemical structure, have been studied for their spasmolytic effects on non-vascular smooth muscle (ileum); **5**, **6b**, **7c**, **7e**, **6g**, **7g**, **6h**, **7h**, and **7l**. **5** and **7c** have no effect in agreement with what observed on vascular smooth muscle. The other tested compounds showed intrinsic activity greater than 50% and interesting potency. In particular **6b**, **7c**, **7e**, **7h**, and **7l** are 117, 71, 121, 7, and 25 times less potent than diltiazem, respectively. Surprisingly, **6g** has a potency comparable to that of diltiazem, while **6h** is 611 times more potent than diltiazem (IC_50_ = 0.00018 µM (c.l. 0.00014–0.00024); IC_50_ = 0.11 µM (c.l. 0.085–0.13) respectively).

### 2.3. Antiproliferative Activity of Compounds **6** and **7**

None of the compounds **6** and **7** showed a significant pharmacological antiproliferative activity against any cell targets (Table 3). In any combination of compounds and cell lines, the IC_50_ were mostly much higher than 30 μM, an arbitrary limit chosen to define a significant pharmacological activity in our experimental conditions. Most importantly, these values differ by 3 orders of magnitude than the mean IC_50_ for doxorubicin (0.01 μM).

Based on the mean curves of human ovarian cancer cell line A2780/DX3, which was selected for doxorubicin, we also calculated the IC_5_ parameters (Table 3). These values were used in the following experiments to evaluate the inhibition of doxorubicin efflux by ABCB1 in these cells.

### 2.4. Determination of Doxorubicin Accumulation by Flow Cytometry

All compounds were then analyzed for their ability to neutralize the efflux of doxorubicin in A2780/DX3 cells. These cells show a multidrug resistance linked to the over-expression of ABCB1 on the membrane (Figure 4).

The flow cytometric analysis results for the reversing activity of compounds **6** and **7**, in resistant A2780/DX3 cells, are summarized in Table 4.

In none of the cases did the co-treatment with our compounds cause a significant pharmacological increase of doxorubicin content in resistant cells, suggesting the absence of inhibition of ABCB1 activity.

It is important to note that in similar experimental conditions, doxorubicin accumulation in A2780 cells (not co-treated with compounds **6** and **7**) was about 4.6 ± 1.3 (460%) times higher than in equally treated resistant A2780/DX3 cells, and that in similar experimental conditions, diltiazem caused a mean increase of doxorubicin accumulation of 699 ± 75% (Standard Error) [14].

## 3. Discussion

Resistance to anticancer therapy is one of the main causes of therapy failure, mostly due to the activation of transport mechanisms responsible for the extrusion of the drug from the cells. Of all the transporters known at the cellular level to be responsible for MDR, Pgps are among the most important [5]. As a matter of fact, many studies have shown a direct correlation between P-glycoprotein (Pgp) increase and resistance to anticancer therapy in various human cancers [23,24,25,26,27]. A wide range of unrelated chemical structures are able to inhibit Pgps [28]. Among these, l-type calcium channel modulators [29,30], used in therapy for cardiovascular diseases, provided this “side effect.” In particular, among the most representative chemotypes—1,4-dihydropyridines, benzothiazepines, and phenylalkylamines—the latter represents the most interesting ones [31,32,33]. The lead compound of the phenylalkylamine class, verapamil, was the first compound to present enormous therapeutic potential as an MDR reverting agent, but its well-known effects on the cardiovascular system prevent its clinical application. This “side effect” of calcium channel modulators cannot be related only to the ability to reduce Ca^2+^ entry into cells, but also to a direct action on Pgps [34]. In fact, verapamil directly inhibits Pgp (PM 170 kDa), while nifedipine with diltiazem binds a 65 kDa protein, thus demonstrating that calcium modulators exert the effects of MDR reverting not only by direct interaction with Pgp, but also by affecting other proteins that can indirectly alter Pgps transport/activity.

Additionally, if the calcium antagonist classes are chemically very heterogeneous, it is possible to identify some common characteristics that are necessary for MDR reverting action: aromatic ring, protonable nitrogen, high lipophilicity, and H-bond interactions. In particular, the lipophilicity is important to help stabilize bonding between drug and Pgps.

Many studies have been devoted to chemical modifications of lead compounds to increase the MDR reverting “side effect” at the expense of cardiovascular effects, therefore finding pharmacophores that are able to relegate the cardiovascular ones to the role of “side effects.” In addition, we have shown that other chemotypes capable of blocking l-type calcium channels also have the same MDR reverting action and that, based on chemical modifications, it is possible to modulate their activity [30]. With this in mind and having long been involved in the study of new chemotypes as calcium channel ligands, we started from this ground to find new MDR reverting candidates [31], demonstrating that appropriate structural changes were requested to increase this action to the detriment of the cardiovascular system. In line with previous studies, we have now examined a new chemotype: 4-nitrothiopyran *S,S*-dioxide (**5**).

In compound **5**, the contemporary presence of two pharmacophores, such as the nitro (several cardiovascular drugs contain the nitrogroup: 1,4-dihydropyridines, such as nifedipine and nicardipine, are just two examples) and the sulphonyl groups, seemed a good premise for opening the way to the discovery of new leads with CEB activity and for evidencing new structure-activity relationships in this field. Biological characterization of **5** has not been able to confirm any significant cardiovascular activity. For this reason, considering that in all of the previously studied compounds [9,10,11,15,35] condensed homo- or hetero-cycles are present, we have turned our attention to the benzocondensed sulfur heterocycles **6** and **7**.

Regarding cardiac parameters, all thiochromans *S,S*-dioxide **6** and **7** are devoid of chronotropic effects, while all derivatives have negative inotropic activity both on the electrically stimulated left atrium and on the spontaneous beating right atrium. In general, 2 phenyl derivatives are more potent than the corresponding 2-H compounds (Figure 5). The most promising pharmacophores for negative inotropic activity are the *p*-methoxyphenyl (**7e**) and the phenyl (**7a**) substituted. None of the studied compounds has spasmolytic effects on vascular smooth muscle. On the contrary, the compounds selected for action on non-vascular smooth muscle all have intrinsic activity greater than 50% except for compound **5**. The spasmolytic activity is ca. 2 times higher for the 3-aryl-4-nitrothiocromans *S,S*-dioxide (**6**), compared to 3-aryl-4-nitro-2-phenylthiochromans *S*,*S*-dioxide (**7**). The action increases if the phenyl in 3 carries an electron-withdrawing group in *meta* or *para* position.

In no case were our selected myocardial l-Type calcium channel modulators **6** and **7** able to significantly inhibit the proliferation of tumor cells, and they did nothing to hamper the ABCB1 activity at low equitoxic concentration.

In fact, compounds **6d** and **6f** were able only to slightly improve doxorubicin accumulation in multidrug resistant A2780/DX3 cells (28% and 24%, respectively), as shown by our determinations by flow cytometry. Although significant on a statistical point of view (*p* < 0.05) our results are not pharmacologically so relevant, our target being an increase of doxorubicin accumulation of at least 400–500% for reaching a comparable rate of doxorubicin accumulation in A2780 cells, the sensitive parental cell line. Thus, it is worth further investigation of **6d** and **6f** in order to enhance their potentiation of intracellular doxorubicin accumulation using molecular docking analysis and drug modeling methods [36], and in order to possibly synthesize new and more active molecules starting from these lead compounds.

In conclusion, on the basis of the preliminary data here reported, we can state that derivatives **6** and **7** are able to modulate l-Type calcium channels, in particular those of non-vascular smooth muscle, while for derivatives of chemotype **6**, there is little selectivity in the MDR reverting action (Figure 6). Appropriate chemical modifications on the scaffold **6** could lead to more interesting results in MDR reverting activities.

## 4. Materials and Methods

### 4.1. Chemistry

The preparation and characterization of thiopyrans *S*,*S*-dioxide **5** [16], and of the thiochromans *S,S*-dioxide **6** and **7** [17,18], has been already reported.

### 4.2. Ex Vivo Studies

The functional profile of compounds **5**, **6**, and **7** was studied on guinea-pig isolated left and right atria to assess their inotropic and/or chronotropic effects, respectively, and on K^+^-depolarized (80 mM) guinea-pig vascular aortic strips to assess calcium antagonist activity. Relevant details have been previously reported [37]. All compounds were checked at increasing concentration to evaluate the percent decrease of developed tension on the isolated left atrium driven at 1 Hz (negative inotropic activity), and the percent decrease in the atrial rate on the spontaneously beating right atrium (negative chronotropic activity). For compounds lacking negative chronotropic effects, the inotropy was also checked in the spontaneously beating right atria. Selected compounds were studied on the K^+^-depolarized ileum smooth muscle strips to evaluate percent inhibition of calcium-induced contraction.

Data were analyzed using Student’s t-test and are presented as mean ± S.E.M. [22]. Since the drugs were added in cumulative manner, the difference between the control and the experimental values at each concentration were tested for a *P* value < 0.05. The potency of drugs defined as EC_50_ was evaluated from log concentration-response curves (Probit analysis using Litchfield and Wilcoxon [37] or GraphPad Prism^®^ software [38,39]) in the appropriate pharmacological preparations. When the compounds were devoid of negative chronotropic activity, the inotropic activity was checked for the spontaneously beating right atria using diltiazem as reference compounds.

Experimental values at each concentration were tested for a *P* value < 0.05. The potency of drugs defined as IC_50_ was evaluated from log concentration-response curves as previously described in Section 2.2.1.

### 4.3. In Vitro Studies

#### 4.3.1. Chemicals

Doxorubicin in clinical formulation was obtained from Pfizer Italia (Milano, Italy) and diluted in normal saline solution to opportune concentrations. All examined compounds **6** (**a**, **d**, **f**) and **7** (**a**, **d**, **h**, **l**), except **7i**, were dissolved in 100% dimethylsulfoxide (DMSO) and then diluted in fetal calf serum, in order to obtain a final 1% DMSO concentration (0.1% on cells). **7i** was dissolved to a final 3% DMSO concentration (0.3% on cells).

#### 4.3.2. Cell Lines

Tumor cell lines SHSY5Y (neuroblastoma), and A2780 (ovary) sensitive to doxorubicin and multidrug-resistant A2780/DX3 cells (provided by Dr. Y.M. Rustum and obtained with exposure to increasing concentrations of doxorubicin), were cultured in an RPMI 1640 medium plus 10% fetal calf serum, 1% glutamine, and 1% penicillin-streptomycin (complete medium). Resistant cells were maintained in a presence of doxorubicin 0.1 μM, removed 3–6 days before experiments. The MDA-MB-453 (breast) cell line was cultured in DMEM medium containing 10% fetal calf serum, 1% glutamine, 1% non-essential amino-acids, and 1% penicillin-streptomycin (complete medium).

#### 4.3.3. MTT Assay

Cell lines were plated at opportune densities/well in 96-well microtiter plates for 6–8 h. To evaluate the concentrations of compounds **6** and **7** that were able to inhibit 5% and 50% (IC_5_ and IC_50_) cell growth, they were administered in 5 different concentrations (10-fold serial dilutions, starting from the maximum concentration of 100 μM); the maximal final volume/well was 200 μL. After three days, 50 μL of MTT (Sigma, St. Louis, MO, USA) solution (2 mg/mL in PBS) was added to each well, and microtiter plates were incubated at 37 °C for 4 h. Thereafter, the culture medium was carefully aspirated; 100 μL of 100% DMSO was added and incubated for 20 min, and a complete and homogeneous solubilization of formazan crystals was achieved by shaking the well contents.

The absorbance was measured with a Microplate Reader iMark (Bio-Rad Laboratories, Milano, Italy) at 595 nm. IC_5_s (compound concentration inhibiting 5% cell proliferation) and IC_50_s (compound concentration inhibiting 50% cell proliferation) were calculated with the analysis of single dose response curves basis. Each experiment was repeated four times.

#### 4.3.4. Study of Doxorubicin Intracellular Accumulation

Multidrug-resistant A2780/DX3 cells were plated in 25 cm^2^ flasks at 1.5 × 10^6^ cells/flask in 10 mL. Eighteen hours later, the cells reached about 65–75% confluence and were treated with **6** and **7** compounds (final volume: 10 mL). The concentration used for each compound was five times higher than the specific IC_5_. After 2 h, a small volume of doxorubicin (29 μL) was added to reach a final concentration of 10 μM, and the incubation was prolonged for other 2 h. After removing the medium, cells were washed once with cold phosphate buffered saline (PBS) in a flask, and then were harvested by trypsin at 37 °C for 5 min, and washed again quickly with cold PBS. Pelleted cells were then fixed at 4° C for 24 h with 3.7% paraformaldehyde in PBS containing 2% sucrose. Untreated (negative control) and positive control cells treated without or with doxorubicin were assayed as well.

The intensity of intracellular fluorescence was assayed with flow cytometry (FACScan, BD Biosciences, Milano, Italy) using 488 nm excitation and 575 nm bandpass filter for doxorubicin detection. Values were expressed in arbitrary units as mean fluorescence intensity (MFI), while doxorubicin accumulation was calculated as percentage of positive controls as follow (Equation (1)):(MFI treated − MFI negative control)/MFI positive control − MFI negative control.(1)

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
