# Peer review of "3-Aryl-4-nitrobenzothiochromans S,S-dioxide: From Calcium-Channel Modulators Properties to Multidrug-Resistance Reverting Activity"

_molecules, 2020, doi:10.3390/molecules25051056_

Round 1

Reviewer 1 Report

major issues

the introduction in unfocused (the authors jumped between Pgp and L-type Ca channels) and incomplete (especially the paragraph "Our reserach group has been involved ..." should be more informative. the authors described a series of new compounds yet did not provide any spectroscopic data (NMR, HR-MS). Why? I understand that synthetic routes are rather straightforward but the publication of new compounds must be accompanied by NMR, yield, etc.. a lack of statistical analysis of results (apply particularly to Tabl 1 and 4). Without a proper analysis (and the proper positive/negative controls) it is impossible to conclude about the differences in compounds activity a lack of SD in Table 3. In my opinion IC5 (5% activity) is a every pooor indicator of compunds' activity, with high SD expected (the dose response curve slope at this range is very low) the authors express some of Table 4 results as significant (line 230) yet no statistical analysis is present Why DTZ wasn't used for in cellulo studies? Especially studies with Doxo accumulation without DTZ as control doesn't make sense (please, refer line 75) the doxorubicin accumulation is not just efflux, same as drug-resistance. If the authors wanted to analyze the new compounds influence on Pgp pumps, an in vitro test (with pure pgp) would be much better.

minor issues:

replace conf lim with CI in Table 1 line 230 " Interestingly, however, ..."

Author Response

Major issues:

The introduction in unfocused (the authors jumped between Pgp and L-type Ca channels) and incomplete (especially the paragraph "Our reserach group has been involved ..." should be more informative.

The pharagrafh has been modified as follows:…... ”It has long been known that calcium channels might be targets for medicinal chemistry application beyond the cardiovascular system. Unfortunately the most used calcium antagonist have poor selectivity and, as a result, poor application. In this frame, our research group has been involved for a long time in a research program aimed at finding new L-Type calcium channel modulators for peripheral or central application [7,8] by modifying the chemical structure to select alternative action to cardiovascular system.”

The authors described a series of new compounds yet did not provide any spectroscopic data (NMR, HR-MS). Why? I understand that synthetic routes are rather straightforward but the publication of new compounds must be accompanied by NMR, yield, etc..

As reported in Chemistry paragraph (lines 100-102 in the revised file) the products 5-7 are already known (ref 17,18). See also in Materials and Methods paragraph (lines 339-340 in the revised file): ….. “The preparation and characterization of thiopyrans S,S-dioxide 5 [16], and of the thiochromans S,S-dioxide 6 and 7 [17,18] has been already reported.”

A lack of statistical analysis of results (apply particularly to Tabl 1 and 4).

All of the obtained results have been commented.

Without a proper analysis (and the proper positive/negative controls) it is impossible to conclude about the differences in compounds activity a lack of SD in Table 3.

The value that better and simply describes the antiproliferative activity of a compound is its IC50. In our case, in all combinations of molecules and cell lines this parameter was always much higher than 30 μM, a value considered in our experimental conditions the limit for indicating a molecule active in vitro as anticancer compound.

To make clearer this point we rewrote, as also suggested by referee 2, the section "Antiproliferative activity of compounds 6 and 7".

In my opinion IC5 (5% activity) is a every pooor indicator of compunds' activity, with high SD expected (the dose response curve slope at this range is very low).

The IC5s were calculated for each compound on the mean curve obtained from four single experiments. No significant differences were in fact found for the same parameter calculated as the mean of single IC5s elaborated from each experiment. This is the reason of the lack of a significant SD. These concentrations, evident expression of a low activity, were then used for studying the inhibition of doxorubicin efflux from resistant A2780/DX3 cells. T in order to decrease the possible cytotoxic effect of higher concentrations, which could possibly alter the accumulation/efflux values of doxorubicin into the cells also considering the short time of the experiments. In addition the IC5 has been previously published in paper of reference 14: [Viale, M.; Cordazzo, C.; Cosimelli, B.; de Totero, D.; Castagnola, P.; Aiello, C.; Severi, E.; Petrillo G, Cianfriglia, M.; Spinelli, D. Inhibition of MDR1 activity in vitro by a novel class of diltiazem analogues: toward new candidates. J. Med. Chem. 2009, 52, 259–266.]

The authors express some of Table 4 results as significant (line 230) yet no statistical analysis is present.

No statistical activity was performed. The sentence: “a As compared to A2780/DX3 cells treated with doxorubicin alone taken as 100%” simply means that the MFI values for cells treated with 10 mM doxorubicin plus the IC5 of the different molecules was related to the same value obtained in control cells treated only with doxorubicin and here taken as 100%.

To maker clearer the sentence the term “…compared… “was changed with “…related…”.

Why DTZ wasn't used for in cellulo studies? Especially studies with Doxo accumulation without DTZ as control doesn't make sense (please, refer line 75) the doxorubicin accumulation is not just efflux, same as drug-resistance.

We already studied DTZ and published the results of its activity on doxorubicin efflux in A2780/DX3 cells (ref. 14: Viale, M.; Cordazzo, C.; Cosimelli, B.; de Totero, D.; Castagnola, P.; Aiello, C.; Severi, E.; Petrillo G, Cianfriglia, M.; Spinelli, D. Inhibition of MDR1 activity in vitro by a novel class of diltiazem analogues: toward new candidates. J. Med. Chem. 2009, 52, 259–266.). Here it was not necessary to consider a positive control since all our results demonstrated no pharmacologically significant doxorubicin accumulation when cells were co-treated with our compounds. In any case we cited our paper and data about DTZ in the text (lines 243-244, see after).

If the authors wanted to analyze the new compounds influence on Pgp pumps, an in vitro test (with pure pgp) would be much better.

The cytofluorimetric cellular assay used in our experiments is probably the most rapid, affordable, cheap, and simple for determining the doxorubicin accumulation linked to the presence of Pgp170 inhibitors. In analogy with other past researches we preferred to use the same assay in order to compare, in case, our results with those already obtained with other diltiazem-like compounds (ref. 14: Viale, M.; Cordazzo, C.; Cosimelli, B.; de Totero, D.; Castagnola, P.; Aiello, C.; Severi, E.; Petrillo G, Cianfriglia, M.; Spinelli, D. Inhibition of MDR1 activity in vitro by a novel class of diltiazem analogues: toward new candidates. J. Med. Chem. 2009, 52, 259–266.).

Minor issues:

Replace conf lim with CI in Table 1.

We replace “conf lim” with “CI” in Table 1 and in Table 2 as required.

Line 230 " Interestingly, however, ..."

Finally, we erased the text lines 234-237 and added the sentence “…and that in similar experimental conditions Diltiazem caused a mean percent increase accumulation of doxorubicin of 699 ± 75% (Standard Error, ref.14: Viale, M.; Cordazzo, C.; Cosimelli, B.; de Totero, D.; Castagnola, P.; Aiello, C.; Severi, E.; Petrillo G, Cianfriglia, M.; Spinelli, D. Inhibition of MDR1 activity in vitro by a novel class of diltiazem analogues: toward new candidates. J. Med. Chem. 2009, 52, 259–266.).” at lines 243-244. Furthermore, we changed the words “…lead to interesting …” to “…lead to more interesting…” at line 330.

Reviewer 2 Report

In general the presented manuscript is very interesting.

Overall, the manuscript is generally clearly written, and the figures are clear.

The detailed comments are presented below:

You should detail the abbreviation, when you use them for the first time.

The English written style should be revised by a English native speaker and the language requires some reconsideration in order to remove grammar and spelling inaccuracies and to make the manuscript more formal.

The phrase is extremely short, some passage can be rephrased.

The "Antiproliferative activity of compounds 6 and 7"  can be rewrite, it's not clear explain!

The conclusion need to be clear and short!

Thanks for opportunity of reading the review.

Author Response

Comments and Suggestions for Authors

In general the presented manuscript is very interesting. Overall, the manuscript is generally clearly written, and the figures are clear.

The detailed comments are presented below:

You should detail the abbreviation, when you use them for the first time.

The English written style should be revised by a English native speaker and the language requires some reconsideration in order to remove grammar and spelling inaccuracies and to make the manuscript more formal.

We checked the paper and corrected the grammar mistakes.

The phrase is extremely short, some passage can be rephrased.

The "Antiproliferative activity of compounds 6 and 7"can be rewrite, it's not clear explain!

As suggested by the referee the section: "Antiproliferative activity of compounds 6 and 7" has been rewrote as follow:……..”None of compounds 6 and 7 showed a significant pharmacological antiproliferative activity against any cell target (Table 3). In any combination of compounds and cell lines the IC50 were mostly much higher than 30 μM, an arbitrary limit chosen to define a significant pharmacological activity in our experimental conditions. Most importantly, these values differ of 3 times of order of magnetude than the mean IC50 for doxorubicin (0.01 μM). Based on the mean curves of A2780/DX3 cell treatments, we also calculated the IC5 parameters. These values were used in the following experiments to evaluate the inhibition of doxorubicin efflux by ABCB1 in A2780/DX3 cells.

The conclusion need to be clear and short!

In line with your requirements, we are working to the syntheses of modified compounds in order to obtain further biological data.

Reviewer 3 Report

The paper under review presents interesting cardiovascular activities of 3-aryl-4-nitrobenzothiochromans S,S-dioxide.

In my opinion the paper is worth studying and the manuscript contains enough original and interesting material. It is clearly and concisely written. The experimental procedures are described comprehensively. The results are interesting.

I suggest removing the "Chemistry" section from the manuscript, as it seems unnecessary.

Author Response

The paper under review presents interesting cardiovascular activities of 3-aryl-4-nitrobenzothiochromans S,S-dioxide.

In my opinion the paper is worth studying and the manuscript contains enough original and interesting material. It is clearly and concisely written. The experimental procedures are described comprehensively. The results are interesting.

We thank referee for his kind comments.

I suggest removing the "Chemistry" section from the manuscript, as it seems unnecessary.

We think that the Chemistry section as it is, is necessary to open the way to our further investigations

Round 2

Reviewer 2 Report

The authors carried out an extensive revision of the manuscript. 

They improved the language usage, organization and clarity. 

I believe this revision satisfactorily address the concerns I previously raised.

The Authors have correctly amended the manuscript and updated the manuscript to a suitably clarified and proper version.

Author Response

Comments and Suggestions for Authors

The authors carried out an extensive revision of the manuscript.

They improved the language usage, organization and clarity.

I believe this revision satisfactorily address the concerns I previously raised.

The Authors have correctly amended the manuscript and updated the manuscript to a suitably clarified and proper version.

We thank the referee for appreciating our review. In this second review, we focused on English editing.
